# Overcoming Multidrug Resistance Using DNA-Localized Auger Emitters: A Comparative Analysis of Radiotoxicity in Breast Cancer Cells

**DOI:** 10.3390/ijms26135958

**Published:** 2025-06-20

**Authors:** Klaus Schomäcker, Beate Zimmermanns, Thomas Fischer, Markus Dietlein, Ferdinand Sudbrock, Feodor Braun, Felix Dietlein, Melanie von Brandenstein, Alexander Drzezga

**Affiliations:** 1Department of Nuclear Medicine, Faculty of Medicine and University Hospital Cologne, University of Cologne, Kerpener Str. 62, 50937 Cologne, Germany; beate.zimmermanns@uk-koeln.de (B.Z.); thomas.fischer@uk-koeln.de (T.F.); markus.dietlein@uk-koeln.de (M.D.); ferdinand.sudbrock@uk-koeln.de (F.S.); feodor.braun@gmail.com (F.B.); alexander.drzezga@uk-koeln.de (A.D.); 2Computational Health Informatics Program, Boston Children’s Hospital, Harvard Medical School, Boston, MA 02115, USA; felix.dietlein@childrens.harvard.edu; 3Department of Urology, Faculty of Medicine and University Hospital of Cologne, University of Cologne, 50937 Cologne, Germany; melanie.freifrau-von-brandenstein@uk-koeln.de; 4Forschungszentrum Jülich GmbH, Institute of Neuroscience and Medicine, Nuclear Chemistry (INM-5), Wilhelm-Johnen-Straße, 52428 Jülich, Germany; 5German Center for Neurodegenerative Diseases (DZNE), Bonn-Cologne, Venusberg-Campus 1/99, 53127 Bonn, Germany

**Keywords:** Auger electron emitters, multidrug resistance (MDR), DNA-targeted radiopharmaceuticals, breast cancer cell lines, [^125^I]I-iododeoxyuridine (IdU)

## Abstract

Multidrug resistance (MDR) represents a major obstacle to successful chemotherapy and, due to overlapping defense mechanisms, such as enhanced DNA repair and the evasion of apoptosis, can also be associated with radioresistance. In this study, we investigated whether MDR breast cancer cells (MCF-7/CMF) exhibit reduced susceptibility to radiation-induced DNA fragmentation compared to their non-resistant parental counterpart (MCF-7). Using a nucleosome-based ELISA, we quantified the chromatin fragmentation in MCF-7 and MCF-7/CMF cells following their exposure to four radiopharmaceuticals: [^99m^Tc]pertechnetate, [^131^I]NaI (sodium iodide), [^125^I]NaI, and the DNA-incorporating compound [^125^I]iododeoxyuridine ([^125^I]IdU). Each radioactive preparation was assessed across a range of activity concentrations, using a two-way ANOVA. For [^99m^Tc]pertechnetate and [^131^I]NaI, significantly higher DNA fragmentation was observed in the sensitive cell line, whereas [^125^I]NaI showed no significant difference between the two phenotypes. In contrast to the other radiopharmaceuticals, [^125^I]IdU induced greater fragmentation in resistant cells. This finding was supported by the statistical analysis (a 63.7% increase) and visualized in the corresponding dose–response plots. These results highlight the critical role of the intranuclear enrichment of Auger emitters and support further development of radiopharmaceuticals in accordance with this principle. Our data suggest that radiotoxicity is governed not by linear energy transfer (LET) alone, but, fundamentally, by the spatial proximity of the radionuclide to the DNA. Targeting tumor cell DNA with precision radiotherapeutics may, therefore, offer a rational strategy to overcome MDR in breast cancer.

## 1. Introduction

Multidrug resistance (MDR) remains one of the major obstacles in the treatment of advanced breast cancer. Across all subtypes, including hormone receptor-positive, HER2-overexpressing, and triple-negative tumors, many malignancies eventually develop resistance to structurally and mechanistically diverse chemotherapeutic agents. MDR arises through converging defense mechanisms, such as increased drug efflux (e.g., ABCB1, ABCC1), enhanced DNA repair, the evasion of apoptosis, and the emergence of stem cell-like phenotypes [1,2,3].

Emerging evidence suggests that MDR cells often exhibit cross-resistance to ionizing radiation as well, largely due to overlapping cellular mechanisms, such as improved DNA repair capacity and reduced apoptotic responsiveness [4,5,6]. This dual resistance severely limits the therapeutic time window of conventional regimens and underscores the need for approaches that can bypass or directly overcome both chemotherapy and radiotherapy resistance.

Although agents such as eribulin [7] and sacituzumab govitecan [8] are used in heavily pretreated metastatic breast cancer patients, often exhibiting multidrug resistance (MDR), their clinical efficacy does not stem from the direct targeting of MDR mechanisms. Instead, these drugs demonstrate residual activity despite the presence of MDR, but they do not reverse resistance.

Eribulin, a microtubule dynamics inhibitor, is known to be a substrate of MDR transporters (e.g., P-glycoprotein/ABCB1 and ABCC11), and its activity is significantly reduced in MDR cell lines overexpressing these pumps [9].

Sacituzumab govitecan, an antibody–drug conjugate targeting Trop-2, has demonstrated promising clinical activity in heavily pretreated patients with triple-negative breast cancer (TNBC), a subgroup often characterized by therapy resistance [10]. Its mechanism relies on delivering the cytotoxic payload SN-38 to Trop-2-expressing tumor cells, thereby inducing apoptosis, even in chemoresistant lesions [11,12]. However, the applicability of this approach remains limited to tumors with sufficient Trop-2 expression, and its efficacy in broader MDR contexts, especially in luminal breast cancer models lacking Trop-2, such as MCF-7, has not been established. Moreover, systemic toxicities, including neutropenia and gastrointestinal side effects, may compromise tolerability in fragile patient populations.

Among the natural compounds investigated for their ability to overcome multidrug resistance (MDR), resveratrol has been shown to reverse doxorubicin resistance in breast cancer cell models. In MCF-7/ADR cells, resveratrol significantly downregulated the expression of the efflux transporter P-glycoprotein (ABCB1) at both mRNA and protein levels, thereby increasing the intracellular accumulation of doxorubicin. These effects were mechanistically linked to the inhibition of the PI3K/Akt signaling pathway, which is known to regulate drug efflux and survival in resistant cancer cells [10]. In addition to resveratrol, other phytochemicals, including curcumin, epigallocatechin gallate (EGCG), quercetin, and silibinin, have demonstrated potential to modulate ABC transporter activity or alter redox and signaling pathways associated with MDR phenotypes. These agents may act synergistically with chemotherapeutics, interfere with cancer stem cell-related resistance mechanisms, or re-sensitize tumor cells by targeting survival and detoxification networks [13,14].

Valproic acid (VPA), a histone deacetylase inhibitor, has been shown to partially reverse multidrug resistance (MDR) in breast cancer by downregulating the expression of efflux transporters, such as P-glycoprotein (P-gp, encoded by the MDR1 gene), and modulating key survival pathways, including PI3K/AKT signaling. In resistant MCF-7/ADR cells, VPA restored chemosensitivity to doxorubicin and reduced the MDR phenotype at the epigenetic level, highlighting its potential as an adjuvant strategy to counteract chemoresistance mechanisms in breast cancer [15].

The breast cancer resistance protein (BCRP/ABCG2), a member of the ATP-binding cassette (ABC) transporter family, plays a pivotal role in mediating multidrug resistance by actively effluxing a broad range of chemotherapeutic agents out of cancer cells. Targeting BCRP has emerged as a promising strategy to overcome MDR in breast cancer. Numerous small-molecule inhibitors, such as Ko143, fumitremorgin C analogs, and tyrosine kinase inhibitors, have been investigated to modulate BCRP function and restore drug sensitivity. However, clinical translation remains limited due to off-target effects, toxicity, and pharmacokinetic challenges. Thus, while the BCRP remains a compelling molecular target, more selective and safer approaches are required for effective MDR reversal in breast cancer patients [16].

Recent advancements in nanotechnology have enabled the development of nano-drug delivery systems (NDDSs) as a promising strategy to overcome multidrug resistance (MDR) in breast cancer. These systems enhance intracellular drug accumulation by bypassing efflux pumps, such as P-glycoprotein, and improve drug bioavailability at the tumor site via passive and active targeting mechanisms. Liposomes, polymeric nanoparticles, micelles, and dendrimers have all shown preclinical potential in regard to reversing MDR by facilitating drug delivery into resistant tumor cells, while minimizing systemic toxicity. Nonetheless, despite encouraging in vitro and in vivo data, most NDDS approaches remain in early translational stages, and clinical validation is still pending [2,17].

In addition to phytochemicals and natural compounds, hydroxamic acid-based histone deacetylase inhibitors (HDACis), such as vorinostat and panobinostat, have also shown promise in reversing MDR phenotypes. By downregulating key efflux transporters like P-glycoprotein (ABCB1) and BCRP (ABCG2) and modifying chromatin accessibility, HDACis may enhance intracellular drug retention and re-sensitize resistant breast cancer cells to conventional chemotherapeutics [18].

Overall, these findings underscore the breadth of the pharmacological and nanotechnological strategies currently under investigation to overcome multidrug resistance (MDR) in breast cancer, from the modulation of cellular efflux mechanisms and epigenetic reprogramming to advanced drug delivery systems. However, most of these approaches act primarily through indirect mechanisms, demonstrate limited efficacy in clinically resistant tumor phenotypes, or remain in the preclinical stages of development. This highlights the rationale for advancing therapeutic strategies that do not merely circumvent resistance, but aim to overcome it entirely, by directly targeting the nuclear control center of the tumor cell, its DNA. Radionuclides capable of emitting Auger electrons offer a highly localized means of inducing lethal DNA damage when delivered to the cell nucleus [19,20,21], thereby bypassing conventional resistance pathways.

Iodine-125, when stably incorporated into nuclear DNA, represents a promising candidate for circumventing both chemoresistance and radioresistance by delivering spatially confined, high-LET Auger electron emissions directly to the genomic substrate.

Auger electron emitters (AEEs), such as iodine-125, release high-linear energy transfer (LET) electrons over submicrometer distances, producing densely ionizing clusters and complex DNA damage in the immediate vicinity of the decay sites [22,23].

A prototypical example of this concept is [^125^I]I-iododeoxyuridine ([^125^I]IdU), a thymidine analog that becomes incorporated into replicating DNA and delivers cytotoxic Auger effects precisely at the site of genomic replication [24]. Although not suitable for clinical use due to its non-selective incorporation into dividing healthy cells, [^125^I]IdU serves as a powerful proof-of-concept molecule for evaluating the therapeutic potential of DNA-targeted Auger radiation in MDR models.

In this study, we investigated the response of chemotherapy-sensitive and multidrug-resistant (MDR) breast cancer cell lines to a panel of radioactive compounds, ranging from low-LET external emitters to DNA-incorporating Auger electron emitters (AEEs), to assess whether the spatial proximity to nuclear DNA serves as a decisive determinant of cytotoxic efficacy, independent of resistance mechanisms. To quantify DNA fragmentation, we used the Cell Death Detection ELISA, a highly sensitive immunoassay that detects mono- and oligonucleosomes, released during cell death. The assay primarily captures intracellular nucleosome release, which may result from apoptosis or direct radiation-induced chromatin disruption [25,26].

Given that nuclear DNA is the principal target of Auger electrons, this method provides a biologically meaningful and technically robust endpoint for comparing radiotoxic effects in resistant and non-resistant tumor cells.

Apoptosis represents a fundamental fail-safe mechanism for eliminating compromised cells; its suppression is a key feature of MDR and contributes to therapy failure. Ionizing radiation, however, has the potential to circumvent MDR by triggering DNA fragmentation directly, bypassing the need for programmed cell death. Although our assay does not distinguish between apoptotic and non-apoptotic mechanisms of fragmentation, it allows for a consistent and sensitive assessment of total chromatin disruption. We, therefore, refer to the measured outcome as cellular DNA fragmentation.

The central aim of this study was to determine whether MDR breast cancer cells exhibit reduced radiosensitivity compared to their non-resistant counterparts, and whether this potential cross-resistance can be mitigated through the nuclear targeting of Auger emitters. By systematically comparing different radionuclide properties and subcellular localizations, we sought to explore the radiobiological vulnerabilities that might be exploited to overcome MDR.

## 2. Results

We present the results for the different radioactive compounds evaluated in this study, including [^99m^Tc]pertechnetate, a pure gamma emitter with a negligible conversion electron component, [^131^I]NaI as a beta emitter, and [^125^I]iodine, in two distinct forms: as free iodide ([^125^I]NaI) and as iododeoxyuridine ([^125^I]IdU). Iodine-125, as an Auger electron emitter, is capable of inducing DNA damage due to densely ionizing radiation when incorporated into nuclear DNA. These agents were assessed for their ability to induce cellular DNA fragmentation in chemotherapy-sensitive and multidrug-resistant breast cancer cells, with particular emphasis on the impact of LET characteristics and subcellular localization on cytotoxic efficacy.

### 2.1. [^99m^Tc] TcO₄^−^ (Pertechnetate)

A highly significant effect of both the cell line and radioactivity concentration on cellular DNA fragmentation was observed (two-way ANOVA: *p* < 0.0001 for both factors), indicating that chemoresistance correlates with radioresistance. A significant interaction between these two variables (*p* = 0.0159) further suggests that the dose–response relationship differs markedly between the cell types. The predicted mean DNA fragmentation level in the drug-sensitive cells was 9.045, compared to only 1.534 in the multidrug-resistant (MDR) cells, a pronounced difference of 7.512 (95% CI: 5.405 to 9.618). This indicates a substantially higher susceptibility to [^99m^Tc]pertechnetate-induced DNA fragmentation in the sensitive cell line.

This pattern is clearly reflected in Figure 1, according to which the blue box plots represent the sensitive cells, and the gray box plots the MDR cells. With increasing activity concentrations (ranging from 0.1 to 500 MBq/mL), the blue boxes rise steeply, demonstrating a marked, dose-dependent increase in DNA fragmentation.

In contrast, the gray boxes remain consistently low and nearly flat across the entire dose range, underscoring the pronounced radioresistance of the MDR cells. The greatest divergence appears at 100 MBq/mL and above, where the blue plots show both elevated medians and broader variability, hallmarks of robust radiation-induced DNA damage in the sensitive cell population.

### 2.2. [^131^I]NaI

A two-way ANOVA revealed a significant interaction between the cell line and the [^131^I]NaI dose (*p* < 0.0001), indicating that the dose–response relationship differed substantially between chemotherapy-sensitive and multidrug-resistant MCF-7 cells. The main effect of the cell line was also highly significant (*p* = 0.0005), confirming that the overall DNA fragmentation levels differed between the two phenotypes. In addition, the main effect of the radionuclide dose reached strong significance (*p* < 0.0001), supporting a dose-dependent induction of cellular DNA fragmentation.

Quantitatively, the mean fragmentation index across all the doses was 1.224 in regard to the sensitive cells and 0.8627 in regard to the resistant cells, yielding a significant mean difference of 0.3717 (95% CI: 0.1645 to 0.5788). This finding reflects a robust and consistent increase in [^131^I]-induced DNA fragmentation in the sensitive cell line.

This difference is further illustrated by the corresponding dose–response plot (Figure 2) while the sensitive cells display a steep upward trajectory beginning at 1.25 MBq/mL and peaking between 5 and 10 MBq/mL, the resistant cells exhibit a markedly flatter response curve, with minimal increases across the full activity range. The error bars confirm the consistency and reproducibility of this divergence.

Taken together, these results demonstrate that [^131^I]NaI induces significantly greater nuclear DNA fragmentation in chemotherapy-sensitive MCF-7 cells compared to their multidrug-resistant counterparts.

### 2.3. [^125^I]NaI

For the low-energy emitter [^125^I]NaI, which is not incorporated into nuclear DNA, no significant interaction between the cell line and dose was observed (two-way ANOVA: *p* = 0.9976), nor was there a significant main effect of the cell line (*p* = 0.7400). However, the main effect of the radionuclide dose was highly significant (*p* < 0.0001), indicating a consistent, dose-dependent increase in DNA fragmentation across both cell types.

Despite the lack of statistical significance of the cell line effect, the numerical difference between the groups was notable. The predicted mean DNA fragmentation value in the sensitive cells was 1.412, compared to only 0.4890 in the resistant cells, resulting in an absolute difference of 0.9225 (95% CI: 0.6789 to 1.166). This substantial gap suggests a biologically relevant divergence that may have been underestimated by the ANOVA model, potentially due to within-group variability or the model’s conservative assumptions.

This interpretation is further supported by the visual data presented in Figure 3. Across the full activity range (0.0001 to 100 MBq/mL), the box plots representing sensitive cells (blue) consistently show higher median values and broader variability than those for resistant cells (black). In the mid-to-high dose range (10–100 MBq/mL), this difference becomes more pronounced, with sensitive cells exhibiting a clear upward trend in DNA fragmentation, while the resistant cells remain tightly clustered at lower levels. This graphical separation strengthens the impression of a consistent biological disparity, even in the absence of formal statistical confirmation of the cell line factor.

Taken together, the results point to a biologically meaningful, though statistically non-significant, difference in DNA fragmentation between sensitive and resistant cells following exposure to [^125^I]NaI.

### 2.4. [^125^I]I-Iododeoxyuridine [^125^I]IdU

For the low-energy emitter [^125^I]IdU, which, upon incorporation into nuclear DNA, delivers Auger electron emissions in direct proximity to the genome, a two-way ANOVA revealed a statistically significant interaction between the cell line and dose (*p* = 0.0077), indicating that the dose–response relationship differed between chemotherapy-sensitive and multidrug-resistant cells. The main effect of the dose was highly significant (*p* < 0.0001), confirming a robust, dose-dependent increase in DNA fragmentation across both cell types. The main effect of the cell line also reached statistical significance (*p* = 0.0326), indicating a general difference in the fragmentation levels between the two phenotypes.

Unexpectedly, the predicted mean DNA fragmentation value in resistant cells (1.414) exceeded that of sensitive cells (1.229), yielding a negative mean difference of –0.1842 (95% CI: −0.3525 to −0.01590). Although modest in absolute terms, this difference was statistically supported and consistent across the dose range. Notably, the cell line factor alone accounted for 63.71% of the total variance in the dataset, underscoring that the observed difference, while numerically small, was the dominant source of variation in this experiment. These findings suggest that resistant cells may be slightly more susceptible to [^125^I]IdU-induced DNA fragmentation than their sensitive counterparts, an inverse pattern compared to all the other tested radionuclides.

This inversion is clearly reflected in the corresponding Figure 4. Across the entire activity range, resistant cells (black) consistently display higher median values and broader interquartile ranges than sensitive cells (blue), particularly at ≥3 MBq/mL. While both cell types exhibit a dose-dependent increase, the steeper rise in resistant cells aligns with the statistical result, further supporting a reversed sensitivity profile for [^125^I]IdU.

### 2.5. Dose-Dependent Effects Across Radionuclides: Analysis of Row Factors

To evaluate the extent to which the cellular DNA fragmentation differed between the chemotherapy-sensitive and multidrug-resistant MCF-7 cells, we examined the row factor from the two-way ANOVA across all four radionuclide conditions. This parameter represents the main effect of the cell line, independent of the dose, and quantifies the proportion of total variance attributable to phenotypic differences between the two cell types.

Notably, the row factor accounted for the largest share of variance in the [^125^I]IdU dataset (63.71%), indicating a robust and consistent difference between resistant and sensitive cells, with higher fragmentation observed in the resistant cell population. For [^131^I]NaI and [^99m^Tc]pertechnetate, the row factor explained 28.45% and 15.44% of the total variance, respectively, both corresponding to increased DNA fragmentation in sensitive cells. In contrast, [^125^I]NaI yielded a row factor contribution of only 3.01%, aligning with the absence of a statistically significant difference between the two phenotypes under this condition.

## 3. Discussion

### 3.1. DNA Fragmentation in the Context of Multidrug Resistance and Radioresistance

In selecting an appropriate method to assess DNA damage, we intentionally opted for a global DNA fragmentation assay rather than foci-based techniques, such as γH2AX quantification [27].

While γH2AX staining offers high-resolution detection of double-strand break formation, it is technically complex, resource intensive, and, owing to its mechanistic specificity, provides a level of detail beyond what was required for our primary objective. Moreover, recent reviews, including that by Valente et al. [28] emphasize that γH2AX signaling is influenced by multiple endogenous and exogenous factors, such as the cell cycle stage, oxidative stress, metabolic status, and physiological apoptosis. These variables can lead to background signals that, although biologically relevant, may confound the precise attribution of γH2AX intensity to radiation-induced DNA damage.

Our goal was to evaluate and compare the overall radiobiological impact of various radionuclides on MDR and non-MDR breast cancer cells. The selected ELISA-based method enabled efficient, scalable, and reproducible quantification of cumulative DNA fragmentation under standardized conditions, thereby enabling direct comparisons across the cell types and radiochemical treatments.

Furthermore, γH2AX assays have already been widely applied in the context of [^125^I]IdU-based studies, providing a robust mechanistic framework. Our investigation complements this body of work by adopting a broader, outcome-centered perspective.

Taken together, our methodological choice represents a deliberate balance between experimental feasibility and biological relevance, aligned with the comparative and exploratory aims of the present study.

Our findings reveal that multidrug-resistant (MDR) breast cancer cells exhibit markedly reduced susceptibility to radiation-induced DNA fragmentation compared to their chemotherapy-sensitive counterparts. In regard to three out of four tested radionuclides, [⁹⁹ᵐTc]pertechnetate, [^131^I]NaI, and [^125^I]NaI, MCF-7 cells with acquired chemoresistance (MCF-7/CMF) displayed significantly lower levels of chromatin disruption, supporting the hypothesis that cross-resistance mechanisms extend beyond cytostatic agents to include ionizing radiation. These results align with previous evidence indicating that MDR phenotypes frequently feature enhanced DNA repair capacity, elevated antioxidant defenses, and dysregulated apoptotic signaling, hallmarks that collectively confer resistance across multiple therapeutic modalities [4,5,6,27].

The sole exception was [^125^I]iododeoxyuridine ([^125^I]IdU), which induced slightly, but in a statistically significant manner, greater DNA fragmentation in the resistant cells. This reversal is biologically plausible: As an iodinated nucleoside analog, [^125^I]IdU is incorporated into genomic DNA during replication, positioning the Auger emitter in immediate proximity to its most vulnerable target. The pronounced effect of the cell line type (row factor = 63.71%) observed in the ANOVA underscores that DNA-incorporated Auger emitters can overcome resistance phenotypes that would otherwise dampen the cellular response to ionizing radiation.

These findings reinforce a central tenet of radiobiology: The therapeutic efficacy of Auger electron emitters (AEEs) is determined not solely by their high linear energy transfer (LET), but, critically, by their subcellular localization at the DNA level. The stark contrast in biological impact between [^125^I]iodide (non-internalizing, minimal effect) and [^125^I]IdU (nuclear incorporation, maximal effect) highlights the need for molecular vectors capable of delivering AEEs in close proximity to nuclear DNA.

Importantly, attempts to correlate the cellular effects with uptake values would be mechanistically misleading in the case of Na[^131^I], Na[^125^I], or [⁹⁹ᵐTc]pertechnetate, as these radionuclides do not appreciably penetrate the cellular membrane under these experimental conditions. Instead, their emissions exert biological effects solely from the extracellular compartment.

Given their physical properties, specifically, the millimeter-range pathlength of β^−^articles from [^131^I] and the essentially unlimited range of γ-rays emitted by all three isotopes, subcellular localization is biologically irrelevant for these agents. The effective dose is, therefore, governed by the total radioactivity present in the surrounding 1 mL volume, rather than by intracellular accumulation.

Correlating the cellular uptake with the biological outcome in these cases would conflate radiophysical range effects with mechanistic targeting, a relationship that holds true only for intracellularly localized emitters. Such mechanistic relevance emerges exclusively with agents like [^125^I]IdU, where Auger electrons are deposited in immediate proximity to nuclear DNA. Only in this context does uptake become a decisive parameter for radiotoxicity.

### 3.2. Dosimetric Considerations

The pronounced cytotoxic effect of [^125^I]iododeoxyuridine cannot be explained by energy deposition alone. Notably, the radiation dose delivered by [^125^I]NaI in a 1 mL aqueous medium is approximately fivefold lower than that of [^131^I]NaI (0.86 Gy vs. 4.4 Gy). The defining feature of [^125^I]IdU lies in its selective incorporation into nuclear DNA.

This mechanistic precision forms the rationale for classifying Auger electron emitters, such as ^125^I, as HILED (highly localized energy deposition) agents. The essential criterion for this designation is the emitter’s spatial confinement to subcellular structures of critical vulnerability, most notably DNA.

In contrast, [^99m^Tc]pertechnetate, which emits only a small number of low-LET conversion electrons (more than 30-fold fewer than ^125^I), required high activity concentrations to elicit detectable DNA damage, and only in the chemotherapy-sensitive cell line. These conversion electrons, lacking proximity to DNA, contribute minimally, if at all, to radiotoxicity.

Moreover, [^131^I]NaI, a β⁻ emitter with intermediate LET and a longer pathlength, effectively induced DNA fragmentation at substantially lower doses and reliably distinguished between resistant and sensitive cells.

To underscore the mechanistic differences among the tested radionuclides, Table 1 provides a comparative summary of their key radiophysical properties, subcellular localization patterns, and anticipated radiobiological effects. This side-by-side overview reinforces the interpretation that the potent cytotoxicity of [^125^I]IdU depends critically on its nuclear incorporation, whereas [^99m^Tc]pertechnetate and [^131^I]iodide exert their effects predominantly through nonspecific extracellular irradiation.

### 3.3. Theranostic Implications

These findings have important theranostic implications. They underscore that the efficacy of Auger-based radiotherapeutics hinges not solely on radionuclide selection, but, critically, on precise subcellular targeting. Building upon this principle, our group is currently advancing estrogen receptor-directed radioconjugates labeled with AEEs. These agents, already evaluated in preclinical models, are designed to leverage receptor-mediated nuclear trafficking to deliver DNA-proximal radiation in hormone receptor-positive tumors.

In parallel, we are investigating novel peptide-based vectors that target cytoskeletal proteins aberrantly expressed during malignant transformation, such as Vimentin-3, as an alternative strategy for intracellular delivery in receptor-negative or triple-negative breast cancers.

The ability of [^125^I]IdU to induce DNA fragmentation, even in MDR cells, provides a compelling proof of concept that genomic proximity is a decisive factor in overcoming therapy resistance.

While our immunoassay sensitively detects nucleosome-associated DNA fragments, it does not differentiate between direct radiation effects and downstream apoptotic mechanisms. This methodological limitation should be taken into account when interpreting the data. Nevertheless, the assay offers a reliable, quantitative readout of total DNA fragmentation as a surrogate for radiation-induced cytotoxicity.

Collectively, our findings provide a strong rationale for the continued development of DNA-targeted Auger therapeutics as a precision approach to bypass resistance mechanisms and enhance radiotherapeutic efficacy in aggressive breast cancer.

## 4. Materials and Methods

### 4.1. Cells

All cell culture procedures were performed under sterile conditions inside a laminar flow hood (Heraeus Precious Metals GmbH & Co. KG, Heraeusstr. 12–14, 63450 Hanau, Germany). Culture media, phosphate-buffered saline (PBS), and EDTA solutions (PAA LABORATORIES GESELLSCHAFT M.B.H., Haidmannweg 9, AT-4061 Pasching, Austria) were pre-warmed in a water bath. Cells were maintained in a humidified incubator at 37 °C, with 5% CO_2_.

MCF-7 cells (a chemosensitive human breast adenocarcinoma line) and the chemoresistant subline, MCF-7/CMF, were cultured in T-75 flasks, using a standard medium consisting of DMEM, supplemented with 10% fetal bovine serum (FBS) and 1% penicillin/streptomycin (Thermo Fisher Scientific GmbH, Dieselstraße 4, 76227 Karlsruhe, Germany). Both cell lines were kindly provided by the former Center of Advanced European Studies and Research (caesar), now the Max Planck Institute for Neurobiology of Behavior–caesar (MPINB), Bonn, Germany.

To maintain the resistant phenotype, MCF-7/CMF cells were exposed once monthly to a cocktail of cyclophosphamide, methotrexate, and 5-fluorouracil (CMF; Merck KGaA, Frankfurter Straße 250, 64293 Darmstadt, Germany) by adding 10 µL of a 1 mg/mL CMF stock solution to 10 mL of the culture medium. After 4 h of incubation, the cells were washed three times with PBS (PAA LABORATORIES GESELLSCHAFT M.B.H., Haidmannweg 9, AT-4061 Pasching, Austria) and provided with fresh medium.

Cells were subcultured at approximately 80% confluency. Following a PBS wash (calcium and magnesium free), detachment was achieved, using 0.02% EDTA (Sigma-Aldrich Chemie GmbH, Eschenstraße 5, 82024 Taufkirchen bei München, Germany) for 5 min at 37 °C. Detached cells were centrifuged at 1000 rpm for 10 min and resuspended in fresh culture medium.

For the cryopreservation, cell pellets were suspended in a freezing medium (Thermo Fisher Scientific GmbH, Dieselstraße 4, 76227 Karlsruhe, Germany) and stored at −80 °C.

Cell morphology and confluency were routinely assessed via phase-contrast microscopy (Axiovert 25, Carl Zeiss Microscopy GmbH, Carl-Zeiss-Promenade 10, 07745 Jena, Germany), before medium exchange or passaging.

### 4.2. Cell Preparation and Radiopharmaceutical Incubation

Approximately 24 h prior to irradiation, cells were detached, centrifuged, and resuspended in fresh culture medium. Cell concentrations were determined using Fuchs–Rosenthal counting chambers (Assistent^®^; Karl Hecht GmbH & Co. KG, Stettener Strasse 22–24, D-97647 Sondheim vor der Rhön, Germany), following a 1:100 dilution in calcium/magnesium-containing PBS (PAA LABORATORIES GESELLSCHAFT M.B.H., Haidmannweg 9, AT-4061 Pasching, Austria). Based on the resulting counts, the suspensions were adjusted to 1 × 10^5^ cells/mL. Aliquots of 100 µL (1 × 10^4^ cells per well) were then seeded into sterile 96-well microtiter plates (VWR International GmbH, Hilpertstraße 20a, 64295 Darmstadt, Germany). The plates were incubated for 24 h at 37 °C to allow cell attachment prior to radiopharmaceutical exposure.

For each experiment, defined activity concentrations of the respective radionuclides were freshly prepared in the medium and added to the pre-seeded wells at a 1:1 ratio (100 µL radioactive medium + 100 µL existing culture medium), resulting in the final activity concentrations that were calculated, accordingly. Negative control wells containing the medium only were spatially separated during incubation to avoid potential cross-irradiation.

The following radionuclides and activity ranges were employed:

[^99m^Tc]Pertechnetate (γ-emitter, T_1/2_ ≈ 6 h): 0.1–1000 MBq/mL. Due to its low linear energy transfer (LET), higher activity concentrations were required to induce measurable biological effects (Curium Pharma, Alt-Moabit 91d, D-10559 Berlin, Germany).

[^131^I]NaI (β⁻/γ-emitter, T_1/2_ ≈ 8 days): 0.0001–30 MBq/mL. As a classical β⁻ emitter, this radionuclide was expected to cause DNA damage at relatively low activity concentrations (Curium Netherlands B.V., Westerduinweg 3, 1755 LE Petten, The Netherlands).

[^125^I]Iodide (Auger emitter, T_1/2_ ≈ 59 days): 0.0001–100 MBq/mL. Owing to its limited cellular uptake, only modest effects were anticipated despite its high LET (PerkinElmer, Inc., 940 Winter Street, Waltham, MA 02451, USA).

[^125^I]Iododeoxyuridine ([^125^I]IdU): 0.0001–10 MBq/mL. As a DNA-incorporating nucleoside analog, this compound facilitates the direct genomic delivery of Auger radiation (GE Healthcare B.V., De Rondom 8, 5612 AP Eindhoven, The Netherlands).

The broad activity ranges were selected to reflect the distinct energy deposition properties of each nuclide. Since the calculated energy doses for [^99m^Tc] were substantially lower than those for [^131^I], proportionally higher activities were used to ensure biological comparability.

All the irradiation procedures were conducted under sterile conditions, using appropriate lead shielding. The cells were incubated with the radiopharmaceuticals for approximately 43 h, before downstream analyses were performed.

### 4.3. Assessment of Radiation-Induced DNA Fragmentation

Cellular DNA fragmentation was quantified using a commercially available Cell Death Detection ELISA kit (Roche Diagnostics GmbH, Sandhofer Strasse 116, 68305 Mannheim, Germany). This assay detects mono- and oligonucleosomes released from the nucleus, based on a two-site sandwich ELISA, employing a biotinylated anti-histone antibody and a peroxidase-conjugated anti-DNA antibody. Signal detection was performed photometrically via ABTS substrate conversion.

To differentiate between necrosis-associated and apoptosis-associated DNA fragmentation, culture supernatants were first collected to measure the extracellular (necrotic) nucleosome fragments. Subsequently, the remaining adherent cells were lysed, and the intracellular fraction was analyzed to quantify the DNA fragments retained within the cytoplasm. It is important to note that the assay does not distinguish whether nuclear DNA fragmentation occurred as a direct consequence of ionizing radiation or as part of apoptosis-related downstream signaling pathways.

All the samples, including the negative and positive controls, were processed using streptavidin-coated 96-well microtiter plates (Thermo Fisher Scientific GmbH, Im Steingrund 4–6, 63303 Dreieich, Germany). Positive controls consisted of purified nucleosome standards, while background controls contained only incubation buffer. Absorbance was measured at 405 nm, with a reference wavelength of 492 nm, using a microplate reader (Tecan Deutschland GmbH, Werner-von-Siemens-Strasse 23, 74564 Crailsheim, Germany). Optical density (OD), as reported here, is the standard metric denoting absorbance in ELISA-based assays.

DNA fragmentation indices were calculated relative to the untreated controls, using the following formula:Fragmentation [%] = (OD_sample − OD_background)/(OD_control − OD_background) × 100(1)

### 4.4. Statistics

All the data were analyzed using a two-way analysis of variance (ANOVA) to examine the effects of the radionuclide activity (dose factor) and the cell phenotype (sensitive vs. resistant; row factor) on DNA fragmentation. Each radiopharmaceutical, [^99m^Tc]pertechnetate, [^131^I]iodide, [^125^I]iodide, and [^125^I]iododeoxyuridine ([^125^I]IdU), was analyzed using a separate two-way ANOVA model.

Statistical analyses were performed using GraphPad Prism, version 10.4.2 (GraphPad Software, LLC, 2365 Northside Drive, Suite 560, San Diego, CA 92108, USA). Group mean comparisons were conducted without adjustment for multiple testing. A two-tailed *p*-value < 0.05 was considered statistically significant.

Each two-way ANOVA included the following components:

Row factor (dose effect): quantifies the extent to which DNA fragmentation increases across ascending activity concentrations.

Column factor (cell line effect): captures the general difference in fragmentation between sensitive (MCF-7) and resistant (MCF-7/CMF) cells.

Interaction term: determines whether the dose–response relationship varies between the two cell types.

This statistical design allowed us to disentangle three core effects, as follows:(i)whether DNA fragmentation rises with increasing dose;(ii)whether one cell type exhibits higher susceptibility overall;(iii)whether the differential response between cell types depends on the activity concentration.

The DNA fragmentation levels were assessed at ten different activity concentrations (0.05 to 500 MBq/mL), and comparisons were conducted in parallel across both cell lines. To complement the ANOVA, box-and-whisker plots were generated to illustrate the distribution patterns and trends in fragmentation across the dose levels and phenotypes.

For each radionuclide, the estimated marginal means derived from the ANOVA were used to calculate the mean difference in DNA fragmentation between sensitive and resistant cells. Remarkably, a reversal of this difference was observed for [^125^I]IdU, with resistant cells showing slightly higher fragmentation than their sensitive counterparts, a pattern not detected in regard to any of the other tested radionuclides.

## 5. Conclusions

This study shows that multidrug-resistant (MDR) breast cancer cells generally exhibit reduced susceptibility to radiation-induced DNA fragmentation compared to their chemotherapy-sensitive counterparts, a pattern consistent with clinically observed cross-resistance mechanisms. However, the divergent responses to different radionuclides reveal that not all types of ionizing radiation are influenced equally by resistance phenotypes.

Notably, the DNA-incorporated Auger emitter [^125^I]IdU induced significantly greater DNA fragmentation in resistant cells, highlighting the decisive role of subcellular localization. These results demonstrate that the cytotoxic efficacy of Auger electron emitters is determined not solely by their physical emission characteristics, but, critically, by their intracellular delivery and genomic proximity.

By showing that DNA-targeted Auger radiation can overcome resistance, even in MDR cells, this study provides compelling experimental evidence for the rational design of radiopharmaceuticals that combine high linear energy transfer with precise subcellular targeting. Such approaches, for e.g., involving estrogen receptor-guided tracers or ligands targeting malignancy-associated cytoskeletal structures, hold promise for treating tumors that are refractory to both chemotherapy and conventional radiotherapy.

## Figures and Tables

**Figure 1 ijms-26-05958-f001:**
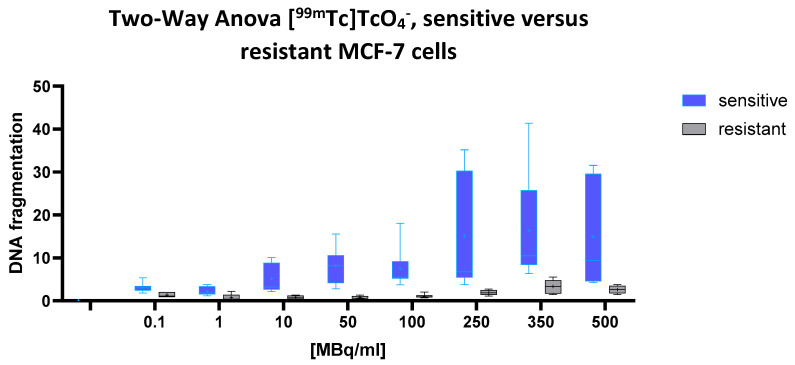
Dose-dependent increase in cellular DNA fragmentation following exposure to [^99m^Tc]TcO₄^−^. DNA fragmentation values (*Y*-axis) represent fold changes in ELISA signal intensity relative to the negative control (untreated cells). These signals reflect the presence of nucleosome-associated chromatin fragments released into the lysate, but do not differentiate between enzymatic cleavage during apoptosis and direct radiation-induced DNA damage. Data represent mean ± SD from n = 8 independent values per condition (each treatment group and cell line combination). Shown are normalized values for chemotherapy-sensitive (blue) and resistant (gray) MCF-7 cells following 43 h of incubation with increasing activity concentrations of [^99m^Tc] TcO₄^−^. Sensitive cells exhibit a marked dose-dependent increase in DNA fragmentation, while resistant cells show minimal response across the full dose range. Box plots indicate median, interquartile range, and full range (whiskers).

**Figure 2 ijms-26-05958-f002:**
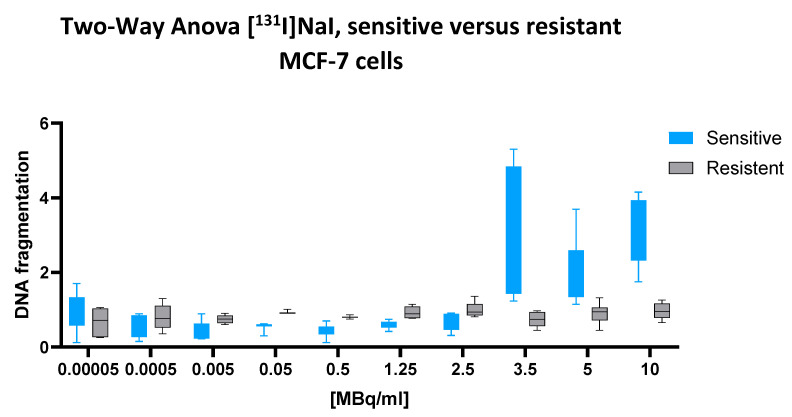
Cellular DNA fragmentation in response to increasing doses of [^131^I]NaI in sensitive and resistant MCF-7 cells. DNA fragmentation values (*Y*-axis) are expressed as fold change relative to the negative control and reflect chromatin-derived nucleosomes released into the cytoplasmic lysate following exposure to ionizing radiation. Data represent mean ± SD from n = 7 independent values per treatment condition and cell line. While the assay cannot distinguish between apoptosis-related cleavage and direct radiation-induced DNA damage, it reliably quantifies overall nuclear fragmentation. After 43 h of incubation with escalating concentrations of [^131^I]NaI, chemotherapy-sensitive cells (blue) exhibit a moderate, dose-dependent increase in DNA fragmentation, which becomes pronounced at higher activity levels. In contrast, resistant cells (gray) remain largely unresponsive throughout the dosage range. Box plots indicate median, interquartile range, and full range (whiskers), illustrating a clear divergence in radiosensitivity between both cell lines at doses ≥ 2.5 MBq/mL.

**Figure 3 ijms-26-05958-f003:**
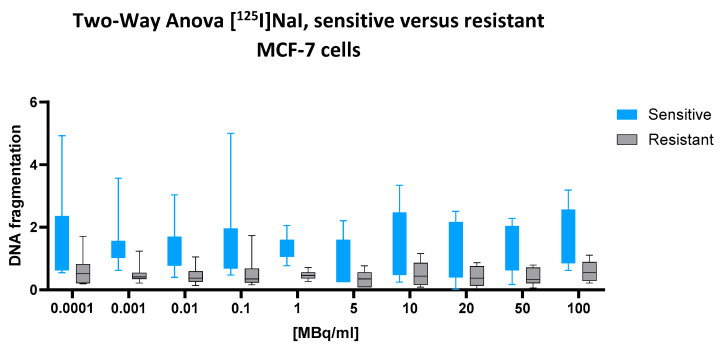
Cellular DNA fragmentation in response to [^125^I]NaI in sensitive and resistant MCF-7 cells. Normalized DNA fragmentation values are presented as fold change relative to the negative control and reflect nuclear DNA damage detected via nucleosome quantification. Data represent mean ± SD from n = 8 independent values per treatment condition and cell line. The assay captures both apoptosis-related and direct radiation-induced fragmentation without distinguishing their origin. Upon incubation with increasing concentrations of [^125^I]NaI for 43 h, chemotherapy-sensitive cells (blue) show a broad but shallow dose–response curve, with elevated median values that remain relatively stable across the dose range. In contrast, resistant cells (gray) exhibit consistently lower values with narrower interquartile ranges. The resulting separation between both cell populations remains modest, highlighting the comparatively limited radiotoxicity of extracellular [^125^I]NaI under these experimental conditions. Box plots represent median, interquartile range, and total range (whiskers).

**Figure 4 ijms-26-05958-f004:**
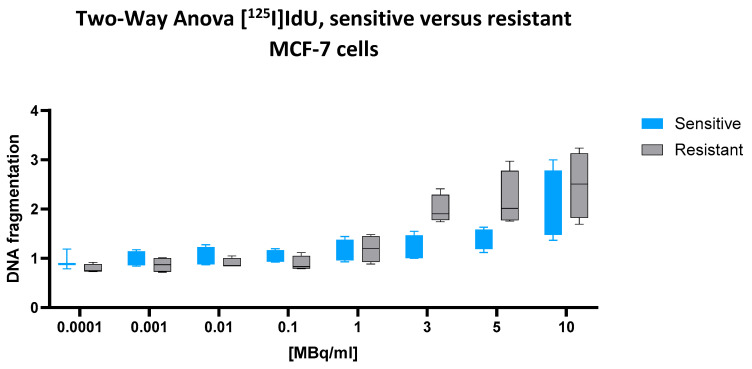
Cellular DNA fragmentation in response to [^125^I]IdU in sensitive and resistant MCF-7 cells. Normalized values represent fold change in DNA fragmentation relative to the negative control. Data represent mean ± SD from n = 4 independent values per treatment condition and cell line. The assay quantifies nucleosome-associated DNA fragments released upon nuclear disintegration, but does not distinguish between direct radiation effects and secondary cellular processes. After 43 h of incubation with increasing concentrations of [^125^I]IdU, resistant cells (gray) consistently show higher fragmentation levels than their sensitive counterparts (blue), especially at higher activity levels (≥3 MBq/mL). This reversal of the typical sensitivity pattern observed with other radionuclides suggests a distinct biological behavior of DNA-incorporated [^125^I]IdU. Box plots indicate the median, interquartile range, and total range (whiskers).

**Table 1 ijms-26-05958-t001:** Radiophysical properties and mechanistic features of the radionuclides used in this study.

Radionuclide	Main Radiation Type	LET	Penetration Range	Cellular Localization	Expected DNA Damage Mechanism
[^125^I]IdU	Auger and conversion electrons	High	Nanometers	Incorporated into DNA	Direct localized DNA breaks
Na[^125^I]	Auger and conversion electrons	High	Nanometers	Extracellular	Negligible; no subcellular targeting
[^99m^Tc]Pertechnetate	γ + few conversion electrons	Low	Centimeters (γ)	Extracellular	Minimal; only at high activity concentrations
Na[^131^I]	β⁻ + γ	Intermediate	~0.6 mm (β⁻), cm (γ)	Extracellular	Generalized DNA damage via medium-based exposure

## Data Availability

The original contributions presented in this study are included in the article. Further inquiries can be directed to the corresponding author.

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
