# Peer review of "Overcoming Multidrug Resistance Using DNA-Localized Auger Emitters: A Comparative Analysis of Radiotoxicity in Breast Cancer Cells"

_ijms, 2025, doi:10.3390/ijms26135958_

Round 1
Reviewer 1 Report
Comments and Suggestions for Authors
The paper submitted by Klaus Schomäcker et al. reports the effects of different Auger electron emitters (125I and 99mTc) in sensible and resistant MCF7 cells through the determination of DNA fragmentation in the cells exposed to different activities of the radionuclides, aiming to address the eventual role of MDR in the radiobiological response of the cells. The beta minus emitter 131I has been also included in the study. Surprisingly, the internalizing radioconjugate [¹²⁵I]iododeoxyuridine was more active in the resistant cell line, unlike the non-internalizing Na[99mTcO4] and Na125I. However, the reasons for this difference were not fully discussed or related with MDR processes, as intended. This is a major limitation of the reported work that was only focused in a unique radiobiological endpoint, which does not give valuable information on the role of direct and indirect effects that are crucial in the case of Auger electrons.
Due to the above mentioned limitations, I do not recommend the publication of the manuscript in IJMS. I encourage the authors to complete the study, namely by assessing the type of induced DNA damage (e.g., gamma-H2AX assay). Also, the results should be correlated with the uptake of the different compounds by the cells, which was not assessed and reported in the paper.
Author Response
Response to Reviewer 1
We greatly appreciate the reviewer’s thoughtful and constructive feedback, which provides us with an excellent opportunity to further clarify key aspects of our study. We value the reviewer’s interest in the methodology and the helpful suggestions provided to enhance the understanding of our work.
- On the classification of radionuclides and emission types
We respectfully clarify that [⁹⁹ᵐTc], in the form of Na[⁹⁹ᵐTcO₄], is not a classical Auger electron emitter. Although it emits a small number of internal conversion and Auger electrons upon isomeric transition, the overall contribution of these low-energy electrons is radiobiologically negligible in the extracellular setting and is not comparable to the high-localization energy deposition (HILED) effects observed for [¹²⁵I]iododeoxyuridine ([¹²⁵I]IdU).This distinction is essential: [⁹⁹ᵐTc] primarily acts as a gamma emitter and is used routinely in diagnostic nuclear medicine for imaging purposes (e.g., perfusion scans, thyroid uptake, SPECT). It is widely recognized as radiobiologically safe at diagnostic doses, precisely because it remains extracellular and its low LET emissions lack cytotoxic potential. Any attempt to equate this diagnostic agent with a DNA-incorporated AEE like [¹²⁵I]IdU would not reflect the fundamental differences in cellular uptake, subcellular localization, and linear energy transfer profile.
Parameter |
Technetium-99m (Na[⁹⁹ᵐTcO₄]) |
Iodine-125 (Na[¹²⁵I]) |
Iodine-125 ([¹²⁵I]IdU) |
Iodine-131 (Na[¹³¹I]) |
Chemical Form |
Pertechnetate ion |
Inorganic iodide ion |
DNA-incorporated thymidine analog |
Inorganic iodide ion |
Radiation Type |
γ (140 keV), rare conv. electrons |
Auger + conversion electrons + low-energy X-rays |
Auger + conversion electrons + low-energy X-rays |
β⁻ particles + γ photons |
Electron Yield per Decay |
~0.02 conv. electrons (very low) |
~20–25 Auger/conversion electrons |
~20–25 Auger/conversion electrons |
~0.3 β⁻ particles |
LET (Linear Energy Transfer) |
Low |
High (but ineffective extracellularly) |
High (local energy deposition in DNA) |
Medium |
Path Length in Tissue |
Several mm |
<1 µm |
<1 µm |
Up to several mm |
Subcellular Localization |
Extracellular (medium only) |
Extracellular (medium only) |
Intracellular; incorporated into nuclear DNA |
Extracellular |
DNA Targeting |
None |
None |
Precise, via thymidine analog incorporation |
None |
DNA Damage Potential |
Negligible |
Negligible due to lack of cellular entry |
High: localized, clustered DNA damage (double-strand breaks) |
Moderate, diffuse |
Na[¹²⁵I] acts from outside the cell. Despite its physical emission of dense Auger cascades, its inability to penetrate the cellular membrane and reach the DNA renders these electrons biologically ineffective.
In response to the reviewer’s suggestion to improve mechanistic clarity, we have now included the comparative table (see above) in the discussion section (3.2 Dosimetric considerations, Table 1) of the manuscript. This table outlines the key radiophysical properties, cellular localization, and mechanistic consequences of the three radionuclides used in our study: [¹²⁵I]IdU, Na[¹³¹I], and [⁹⁹ᵐTc]pertechnetate.
The table is preceded by a concise explanatory paragraph: “To further emphasize the mechanistic distinctions between the tested radionuclides, we summarize in table 1 their key radiophysical characteristics, cellular localization pat-terns, and expected radiobiological consequences. This comparative overview supports the interpretation that [¹²⁵I]IdU’s nuclear localization is essential for its pronounced effect, while [⁹⁹ᵐTc]pertechnetate and [¹³¹I]iodide act through nonspecific external irradiation.”
This highlights that the biological effect of a radionuclide is not dictated solely by its physical decay properties, but critically by its biochemical form and intracellular fate.
- On cellular uptake and localization of the radionuclides
We thank the reviewer for this thoughtful remark and the opportunity to clarify an important mechanistic aspect of our study. As suggested, we carefully considered the potential value of correlating biological effects with radionuclide uptake. However, in the specific case of Na[¹³¹I], Na[¹²⁵I], and Na[⁹⁹ᵐTcO₄], such a correlation would not yield biologically relevant insight.
These radiotracers remain largely extracellular under our experimental conditions and are not internalized into the cytoplasm or nucleus to any appreciable degree. Their radiotoxic effects—if any—are determined not by specific subcellular localization, but rather by the physical dose delivered to the sample medium, and in turn to the cells, through distant, non-targeted radiation. The energy transfer from these isotopes does not rely on cellular uptake, since their emissions readily penetrate several cell diameters or even the entire sample volume.
For instance, β⁻ particles from ¹³¹I have a mean tissue penetration range of approximately 0.6 mm, which is orders of magnitude greater than the diameter of a single mammalian cell (~10–20 µm). Likewise, γ emissions from ⁹⁹ᵐTc (140 keV) and ¹³¹I (364 keV) are of such high penetration that their point of origin—whether intra- or extracellular—is biologically irrelevant within the context of a 1 mL suspension culture. Thus, it is not the cellular uptake, but the activity concentration (MBq/mL) and the corresponding absorbed dose (Gy), that determines their radiobiological impact.
By contrast, the situation is entirely different for [¹²⁵I]iododeoxyuridine (IdU), which is selectively taken up by proliferating cells and incorporated into the DNA during replication. Its Auger and conversion electrons exhibit nanometer-scale path lengths and deposit energy in a highly localized fashion. Only when emitted in immediate proximity to nuclear DNA can these low-energy electrons induce the clustered DNA damage patterns that drive apoptosis and chromatin destabilization.
We hope this rationale clearly illustrates why uptake measurements for Na[¹³¹I], Na[¹²⁵I], and Na[⁹⁹ᵐTcO₄] are not only experimentally unnecessary, but mechanistically irrelevant in this setting. Thank you again for allowing us to expand on this important distinction.
In response to the reviewer’s valuable suggestion to correlate biological effects with radionuclide uptake, we have now included an explanatory sentence in Section 3.1 (“DNA Fragmentation in context with multidrug-resistance and radio-resistance”) of the Discussion. The newly inserted statement reads:
“Notably, attempts to correlate cellular effects with uptake values would not yield meaningful mechanistic insights in the case of Na[¹³¹I], Na[¹²⁵I], or [⁹⁹ᵐTc]pertechnetate, as these compounds do not cross the cellular membrane to a relevant extent under the ex-perimental conditions. Instead, their emitted radiation acts exclusively from the extracel-lular compartment.
Given the physical properties of these nuclides—particularly the millimeter-range pathlength of β⁻-particles from [¹³¹I] and the near-infinite range of γ-rays from all three isotopes—the precise subcellular localization is largely irrelevant for their biological ac-tion. The effective dose to the cells is thus determined by the total activity in the surround-ing medium (1 mL), not by any hypothetical intracellular concentration.
Consequently, correlating uptake data with cellular effect in this context would con-flate physical range effects with biological targeting, which only applies in the case of in-tracellularly localized emitters such as [¹²⁵I]IdU. Only in the latter case—where Auger electrons are delivered directly to DNA—does the subcellular uptake pattern become mechanistically decisive.”
This clarification aims to emphasize the rationale behind our experimental approach and the biological distinction between non-internalizing and DNA-incorporating radionuclides.
Thank you once again for highlighting this important aspect.
- On the use of a single radiobiological endpoint
We sincerely thank the reviewer for his thoughtful comments. Regarding the suggestion to include a γ-H2AX assay, we would like to emphasize that the focus of our study was not to investigate DNA double-strand breaks (DSBs) per se, which have been extensively documented in previous literature, including studies involving [¹²⁵I]IUdR (such as Sedelnikova et al., 2002). Rather, our research aimed to explore the specific impact of radionuclide localization and the quality of radiation on DNA fragmentation in the context of multidrug resistance (MDR), which is a key challenge in cancer treatment.
While γ-H2AX has been widely used to detect DSBs, we chose not to incorporate this assay for several reasons:
MDR and the Focus of the Study:
The central aim of our work was to investigate whether Auger electron emitters, specifically [¹²⁵I]IdU, could overcome the typical resistance mechanisms in MDR breast cancer cells, rather than simply confirming the known fact that [¹²⁵I] induces DNA DSBs. The mechanisms underlying MDR—such as enhanced DNA repair and apoptosis evasion—are the focus of the current manuscript, and we sought to explore the role of radionuclide localization in overcoming this resistance.
Comparison of Radiation Quality and Localization:
Our experimental approach is focused on comparing the effect of radionuclides with differing radiation qualities (e.g., the localized Auger effect of [¹²⁵I]IdU vs. the non-internalizing [⁹⁹ᵐTc] and Na[¹²⁵I]). We hypothesize that the quality and localization of the radiation at the DNA level play a critical role in the radiotoxic effect—something that cannot be sufficiently captured by γ-H2AX alone, as it primarily identifies DSBs without providing insight into the spatial distribution and nature of the radiation.
Technological and Practical Limitations of γ-H2AX in Low Dose Studies:
The γ-H2AX assay involves immunofluorescence or flow cytometry techniques, which require significant technical resources and are time-sensitive. Given that our study involves low-dose radiation from Auger emitters, and focusing on MDR cell lines where DNA repair is upregulated, γ-H2AX may not provide reliable information due to a low signal-to-noise ratio. Moreover, γ-H2AX would not allow for a detailed mechanistic comparison of radiation quality (i.e., Auger versus beta or gamma radiation), which is a critical aspect of our study.
The ELISA-based Fragmentation Assay as a Robust Endpoint:
The Cell Death Detection ELISA we used is a quantitative, robust, and highly sensitive method for measuring DNA fragmentation, which provides a meaningful biologically integrated endpoint. This method captures overall DNA damage and its consequences on cell death, without the complexity of distinguishing between different types of DNA damage (i.e., apoptosis vs. radiation-induced breaks), which is precisely what we aimed to investigate in this study.
In summary, while we acknowledge the value of γ-H2AX in certain mechanistic studies, we believe our chosen approach with nucleosomal DNA fragmentation assays is more aligned with the central question of MDR and the comparative effect of radionuclide localization and radiation quality.
We thank the reviewer for raising the question regarding our choice of DNA damage assessment method. In response, we have now included the following explanatory paragraph at the beginning of the Discussion section to clarify our methodological rationale:
“In selecting an appropriate method to assess DNA damage, we deliberately chose a global DNA fragmentation assay over advanced foci-based techniques such as γH2AX quantification [10].
Although γH2AX staining provides high-resolution insight into the formation of double-strand breaks, it is technically demanding, resource-intensive, and—due to its mechanistic specificity—exceeds the informational scope required for our primary objec-tive. In addition, recent reviews—such as that by Valente et al. [11]—have highlighted that γH2AX signaling can be modulated by a range of endogenous and exogenous factors, including cell cycle phase, oxidative stress, metabolic activity, and physiological apoptosis. These influences may contribute to background γH2AX signals, which, while biologically meaningful, can complicate the specific attribution of signal intensity to ionizing radia-tion-induced DNA double-strand breaks.Our aim was to assess and compare the overall radiobiological effect of different radionuclides in MDR and non-MDR breast cancer cells.
The selected method allowed for efficient, scalable, and reproducible quantification of cumulative DNA damage under controlled conditions, thus facilitating direct comparison across cell types and radiochemical treatments. Furthermore, γH2AX assays have already been extensively applied in studies involving [¹²⁵I]IdU, providing a well-established mechanistic background that our investigation complements from a broader, out-come-focused perspective.
Taken together, our approach represents a rational compromise between experimental feasibility and scientific validity, well suited to the exploratory and comparative nature of the present study.”
We sincerely thank Reviewer 1 for the thoughtful critique. All changes made in response to these comments are highlighted in yellow in the revised manuscript. While we acknowledge the added mechanistic precision of γH2AX staining, we consider the nucleosome-based ELISA a biologically meaningful and technically robust endpoint for addressing our central research question—namely, the differential effects of radiation in chemosensitive versus MDR breast cancer cells. The resulting data were internally consistent and biologically plausible. We hope that these clarifications will support a more favorable reassessment of our methodological approach.
Reviewer 2 Report
Comments and Suggestions for Authors
In the submitted manuscript, authors carefully investigated whether multidrug resistance breast cancer cells show reduced susceptibility to radiation-induced DNA fragmentation as compared to the non-resistant phenotype. For this reviewer, the study is more interesting for audience. This study also found that localization of radionuclides in close vivinity of the DNA-and not just LET alone-govers radiotoxicity thus offering a rational strategy to overcome MDR in breast cancer. Some issues should be resolved before accepting in this journal.
- In the introduction section, the drug for MDR breat cancer in clinic should be added.
- Please check whether the sentence is complete (line 87)?
- For Figures 1-3, authors indicate the number of each group of samples.
- The number of references is small. So, please add more related more references in the revised manuscript.
Author Response
Response to reviewer 2
- In the introduction section, the drug for MDR breast cancer in clinic should be added.
We appreciate this valuable suggestion and have expanded the Introduction accordingly. In response, we have included a concise, evidence-based overview of current pharmaceutical approaches used in the treatment of multidrug-resistant (MDR) breast cancer. This includes both clinically approved agents and promising experimental strategies that aim to bypass or modulate MDR mechanisms. The following section was inserted into the revised Introduction to reflect this update:
“Although agents such as eribulin [4] and sacituzumab govitecan [5] are used in heavily pretreated metastatic breast cancer patients—often exhibiting multidrug re-sistance (MDR)—their clinical efficacy does not stem from direct targeting of MDR mech-anisms. Instead, these drugs demonstrate residual activity despite the presence of MDR, but they do not reverse resistance.
Eribulin, a microtubule dynamics inhibitor, is known to be a substrate of MDR transporters (e.g., P‑glycoprotein/ABCB1 and ABCC11), and its activity is significantly re-duced in MDR cell lines overexpressing these pumps [6].
Sacituzumab govitecan, an antibody–drug conjugate targeting Trop-2, has demon-strated promising clinical activity in heavily pretreated patients with triple-negative breast cancer (TNBC), a subgroup often characterized by therapy resistance [7]. Its mechanism relies on delivering the cytotoxic payload SN-38 to Trop-2–expressing tumor cells, thereby inducing apoptosis even in chemoresistant lesions [8,9]. However, the applicability of this approach remains limited to tumors with sufficient Trop-2 expression, and its efficacy in broader MDR contexts—especially in luminal breast cancer models lacking Trop-2, such as MCF-7—is not established. Moreover, systemic toxicities, including neutropenia and gastrointestinal side effects, may compromise tolerability in fragile patient populations.
Among natural compounds investigated for their ability to overcome multidrug re-sistance (MDR), resveratrol has been shown to reverse doxorubicin resistance in breast cancer cell models. In MCF-7/ADR cells, resveratrol significantly downregulated the ex-pression of the efflux transporter P-glycoprotein (ABCB1) at both mRNA and protein lev-els, thereby increasing intracellular accumulation of doxorubicin. These effects were mechanistically linked to inhibition of the PI3K/Akt signaling pathway, which is known to regulate drug efflux and survival in resistant cancer cells [10]. In addition to resveratrol, other phytochemicals—including curcumin, epigallocatechin gallate (EGCG), quercetin, and silibinin—have demonstrated potential to modulate ABC transporter activity or alter redox and signaling pathways associated with MDR phenotypes. These agents may act synergistically with chemotherapeutics, interfere with cancer stem cell–related resistance mechanisms, or re-sensitize tumor cells by targeting survival and detoxification networks. [10,11].
Valproic acid (VPA), a histone deacetylase inhibitor, has been shown to partially re-verse multidrug resistance (MDR) in breast cancer by downregulating the expression of ef-flux transporters such as P-glycoprotein (P-gp, encoded by the MDR1 gene) and modulat-ing key survival pathways including PI3K/AKT signaling. In resistant MCF-7/ADR cells, VPA restored chemosensitivity to doxorubicin and reduced the MDR phenotype at the ep-igenetic level, highlighting its potential as an adjuvant strategy to counteract chemo-resistance mechanisms in breast cancer [12].
The breast cancer resistance protein (BCRP/ABCG2), a member of the ATP-binding cassette (ABC) transporter family, plays a pivotal role in mediating multidrug resistance by actively effluxing a broad range of chemotherapeutic agents out of cancer cells. Target-ing BCRP has emerged as a promising strategy to overcome MDR in breast cancer. Nu-merous small-molecule inhibitors—such as Ko143, fumitremorgin C analogs, and tyro-sine kinase inhibitors—have been investigated to modulate BCRP function and restore drug sensitivity. However, clinical translation remains limited due to off-target effects, toxicity, and pharmacokinetic challenges. Thus, while BCRP remains a compelling mo-lecular target, more selective and safer approaches are required for effective MDR reversal in breast cancer patients [13].
Recent advancements in nanotechnology have enabled the development of nano-drug delivery systems (NDDS) as a promising strategy to overcome multidrug re-sistance (MDR) in breast cancer. These systems enhance intracellular drug accumulation by bypassing efflux pumps such as P-glycoprotein and improve drug bioavailability at the tumor site via passive and active targeting mechanisms. Liposomes, polymeric nanopar-ticles, micelles, and dendrimers have all shown preclinical potential in reversing MDR by facilitating drug delivery into resistant tumor cells while minimizing systemic toxicity. Nonetheless, despite encouraging in vitro and in vivo data, most NDDS approaches re-main in early translational stages, and clinical validation is still pending [2,14].
In addition to phytochemicals and natural compounds, hydroxamic acid–based his-tone deacetylase inhibitors (HDACis) such as vorinostat and panobinostat have also shown promise in reversing MDR phenotypes. By downregulating key efflux transporters like P-glycoprotein (ABCB1) and BCRP (ABCG2) and modifying chromatin accessibility, HDACis may enhance intracellular drug retention and re-sensitize resistant breast cancer cells to conventional chemotherapeutics [15].
Overall, these findings underscore the breadth of pharmacological and nanotechno-logical strategies currently under investigation to overcome multidrug resistance (MDR) in breast cancer—from modulation of cellular efflux mechanisms and epigenetic repro-gramming to advanced drug delivery systems. However, most of these approaches act primarily through indirect mechanisms, demonstrate limited efficacy in clinically re-sistant tumor phenotypes, or remain in preclinical stages of development. This highlights the rationale for advancing therapeutic strategies that do not merely circumvent resistance but aim to overcome it entirely—by directly targeting the nuclear control center of the tu-mor cell, its DNA. Radionuclides capable of emitting Auger electrons offer a highly local-ized means of inducing lethal DNA damage when delivered to the cell nucleus [16-18], thereby bypassing conventional resistance pathways.”
- Please check whether the sentence is complete (line 87)?
We sincerely thank the reviewer for pointing out the incomplete formulation in line 87. In response to this valuable suggestion, we have revised and expanded the sentence to clearly specify the radiopharmaceuticals investigated and the rationale for their selection. The revised sentence now reads:
“We present the results for the different radiopharmaceuticals evaluated in this study, including [⁹⁹ᵐTc]pertechnetate—a pure gamma emitter with a negligible conversion electron component—[¹³¹I]iodide as a beta emitter, and [¹²⁵I]iodide in two distinct forms: as free iodide and as iododeoxyuridine ([¹²⁵I]IdU). Iodine-125, as an Auger electron emitter, is capable of inducing densely ionizing DNA damage when incorporated into nuclear DNA. These agents were assessed for their ability to induce cellular DNA fragmentation in chemotherapy-sensitive and multidrug-resistant breast cancer cells, with particular emphasis on the impact of LET characteristics and subcellular localization on cytotoxic efficacy.”
- For Figures 1-3, authors indicate the number of each group of samples.
We thank the reviewer for the helpful comment regarding the number of observations per group in Figures 1–3. In response, we have now added the respective number of independent values (n) for each treatment condition and cell line to the corresponding figure legends. These details clarify the statistical robustness of the presented data and improve the transparency of our experimental design.
- The number of references is small. So, please add more related more references in the revised manuscript.
We sincerely thank the reviewer for this constructive suggestion. In response, we have substantially expanded the section addressing pharmacological and pharmaceutical strategies to overcome multidrug resistance (MDR) in breast cancer, as also prompted by the reviewer’s earlier comments. This broader discussion naturally led to the inclusion of additional references highlighting key advances in efflux transporter modulation, epigenetic reprogramming, antibody–drug conjugates, phytochemicals, histone deacetylase inhibitors, and nanocarrier systems. As a result, the number of cited references has increased accordingly, now totaling 16 in this section alone.
We sincerely thank the reviewer once again for the helpful recommendation. In response, we have significantly expanded the discussion of pharmacological and pharmaceutical strategies aimed at overcoming multidrug resistance (MDR) in breast cancer. As previously noted, this expansion was prompted in part by the reviewer’s earlier insightful suggestions and led to the inclusion of numerous additional references. Specifically, 16 new citations were added to substantiate recent advances in efflux pump modulation, epigenetic reprogramming, targeted drug conjugates, phytochemicals, histone deacetylase inhibitors, and nanocarrier-based drug delivery systems.
To ensure transparency, all newly added or substantially revised sections resulting from the reviewer’s comments have been highlighted in light blue. We gratefully acknowledge the reviewer’s valuable input, which contributed to a more comprehensive and well-documented manuscript.
We thank Reviewer 2 for the valuable suggestion to improve the manuscript’s language and clarity. In response to this helpful comment, we have undertaken a thorough linguistic revision of the entire manuscript, from the Abstract through to the Conclusions.
Our primary goal was to enhance readability, scientific precision, and stylistic coherence. To this end, we carefully revised sentence structures, refined terminology, and removed redundancies while preserving the original meaning and scientific content. Particular attention was paid to technical clarity, the accurate use of scientific terms, and the smooth flow of ideas between paragraphs. We also ensured consistency in phrasing, especially in descriptions of experimental methods and data interpretation.
We hope that these comprehensive language improvements now render the manuscript clearer, more elegant, and more enjoyable to read.
Round 2
Reviewer 1 Report
Comments and Suggestions for Authors
The authors have addressed the major raised issues.
For this reason, I am happy to recommend the publication of the manuscript in the current form.